# Multi-enhancer transcriptional hubs confer phenotypic robustness

**Albert Tsai[1†]\*, Mariana RP Alves[1,2†], Justin Crocker[1]\***

[1]European Molecular Biology Laboratory, Heidelberg, Germany; [2]Collaboration for joint PhD degree between EMBL and Heidelberg University, Faculty of Biosciences, Heidelberg, Germany

**Abstract** We previously showed in *Drosophila melanogaster* embryos that low-affinity Ultrabithorax (Ubx)-responsive *shavenbaby* (*svb*) enhancers drive expression using localized transcriptional environments and that active *svb* enhancers on different chromosomes tended to colocalize (Tsai et al., 2017). Here, we test the hypothesis that these multi-enhancer 'hubs' improve phenotypic resilience to stress by buffering against decreases in transcription factor concentrations and transcriptional output. Deleting a redundant enhancer from the *svb* locus led to reduced trichome numbers in embryos raised at elevated temperatures. Using high-resolution fluorescence microscopy, we observed lower Ubx concentration and transcriptional output in this deletion allele. Transcription sites of the full *svb cis*-regulatory region inserted into a different chromosome colocalized with the *svb* locus, increasing Ubx concentration, the transcriptional output of *svb*, and partially rescuing the phenotype. Thus, multiple enhancers could reinforce a local transcriptional hub to buffer against environmental stresses and genetic perturbations, providing a mechanism for phenotypical robustness.

DOI: https://doi.org/10.7554/eLife.45325.001

**\*For correspondence:**
albert.tsai@embl.de (AT);
justin.crocker@embl.de (JC)

[†]These authors contributed equally to this work

**Competing interests:** The authors declare that no competing interests exist.

## Introduction

During embryogenesis, transcriptional regulation controls precise patterns of gene-expression, leading to cell-fate specification (*Long et al., 2016*; *Mallo and Alonso, 2013*; *Reiter et al., 2017*; *Spitz and Furlong, 2012*). This involves coordinating a complex series of interactions between transcription factors and their target binding sites on DNA, leading to the recruitment or exclusion of active RNA polymerases, which determines the transcriptional state of the gene. Live imaging experiments have shown that transcription factor binding in eukaryotic cell lines and embryos is dynamic but transient, occurring frequently but with each event lasting for at most a few seconds (*Chen et al., 2014*; *Izeddin et al., 2014*; *Liu et al., 2014*; *Normanno et al., 2015*). Additionally, recent studies have shown that many developmental enhancers harbor functionally important low-affinity binding sites (*Antosova et al., 2016*; *Crocker et al., 2010*; *Crocker et al., 2015*; *Crocker et al., 2016*; *Farley et al., 2015*; *Farley et al., 2016*; *Gaudet and Mango, 2002*; *Lebrecht et al., 2005*; *Lorberbaum et al., 2016*; *Rister et al., 2015*; *Rowan et al., 2010*; *Tanay, 2006*). One example is the Homeobox (Hox) family that is responsible for body segment identity along the anterior-posterior axis in animals. Because Hox transcription factors descended from a common ancestor, their preferences for binding sequences are very similar (*Berger et al., 2008*; *McGinnis and Krumlauf, 1992*; *Noyes et al., 2008*). To select for specific Hox factors, several enhancers in the *shavenbaby* (*svb*) locus make use of low-affinity binding sequences for Ultrabithorax (Ubx) (*Crocker et al., 2015*). Svb is a transcription factor that drives the formation of trichomes, epidermal projections on the surface of the segmented fly embryo (*Chanut-Delalande et al., 2006*; *Delon et al., 2003*; *Payre et al., 1999*). Thus, a key question was how low affinity binding sequences are able to drive strong transcriptional activation in developing embryos.

We have previously shown in living *Drosophila melanogaster* embryos that Ubx transiently but repeatedly explores the same physical locations in a nucleus, which are likely clusters of binding sites (*Tsai et al., 2017*). We have additionally shown that transcriptional microenvironments of high local Ubx and cofactor concentrations surround active transcription sites driven by low-affinity *svb* enhancers. As the distributions of many transcription factors in the nucleus are highly heterogeneous, the transcriptional activity of low-affinity enhancers would depend on the local microenvironments. Interestingly, we observed that transcriptionally active, minimalized versions of two of the three ventral *svb* enhancers (*E3* and *7*) are Ubx-responsive and preferentially appear near or overlap spatially with transcription sites of the endogenous *svb* gene, despite being on different chromosomes (*Crocker et al., 2015*; *Tsai et al., 2017*). This colocalization suggests microenvironments could be shared between related enhancers to increase transcriptional output, where enhancers would synergistically form a larger local trap for transcription factors than each could alone. Retaining multiple enhancers within a microenvironment could also provide redundancy in case individual enhancers are compromised and buffer negative impacts when the system is subjected to stress. This idea is consistent with the observed phenotypic robustness of *svb* enhancers in maintaining sufficient trichome numbers even under temperature stress (*Crocker et al., 2015*; *Frankel et al., 2010*). These results are consistent with multi-component transcriptional 'hubs' that are local areas enriched for components of the transcriptional machinery and transcription factors through multiple attractive and cooperative interactions (*Boija et al., 2018*; *Cisse et al., 2013*; *Furlong and Levine, 2018*; *Ghavi-Helm et al., 2014*; *Lim et al., 2018*; *Mir et al., 2017*; *Mir et al., 2018*). One potential building block for these 'hubs' are multiple, long-range, enhancer-to-enhancer interactions. However, it is not yet understood how such multivalent interactions function mechanistically, and how they contribute to phenotypic robustness.

To understand the mechanistic implications of having multiple enhancers in a shared microenvironment, here we examined the ability of the *svb* locus to maintain transcriptional output and produce the correct phenotype under temperature-induced stress in flies harboring a deletion of a partially redundant enhancer—the *DG3* ventral enhancer. When embryos were raised at high temperatures, we observed phenotypical defects in ventral trichome formation for the *DG3*-deletion *svb* allele but not for the wild-type. At the molecular level, Ubx concentrations around transcription sites of the *DG3*-deletion allele decreased. The transcriptional output of *svb* without *DG3* also decreased. To test the hypothesis that shared microenvironments modulate transcriptional output and provide buffering under stress, we sought to rescue the *DG3*-deletion allele through inserting the complete *svb cis*-regulatory region on a BAC (*svbBAC*) on a different chromosome. We observed that Ubx concentration around active transcription sites of the *DG3*-deletion allele and their transcriptional output increased when the *svbBAC* is physically nearby. Moreover, we found that trichome formation was partially rescued at high temperature. As a result, our findings support the hypothesis that shared microenvironments provide a mechanism for phenotypic robustness.

## Results

### The *DG3* enhancer responds specifically to Ubx in the A1 segment

The ventral *svb* enhancers *DG3*, *E3* and *7* (*Figure 1A*) contain low-affinity Ubx binding sites and have been shown to be transcribed in microenvironments of high Ubx concentrations in the first abdominal (A1) segment on the ventral surface of the embryo (*Tsai et al., 2017*). Each of these enhancers produces ventral stripes of expression along segments A1-A7 in the embryo, resembling the endogenous expression pattern of *svb* (*Figure 1B*). Each enhancer contributes to different but partially overlapping portions of the total expression pattern. Furthermore, they have different Ubx ChIP enrichment profiles (*Figure 1—figure supplement 1*). Whereas the interaction of *E3* and *7* with Ubx had been previously explored in detail (*Crocker et al., 2015*), *DG3* remained unexplored. Therefore, we tested the response of the *DG3* enhancer to Ubx by altering Ubx levels and measuring the transcriptional output with a reporter gene (*lacZ*). In wild-type embryos, the *DG3* reporter gene was expressed ventrally in stripes along segments A1-A7, in addition to narrow thoracic stripes in T1-T3 (*Figure 1C*). Expression from *DG3* on the ventral surface in the T2 and T3 segments was

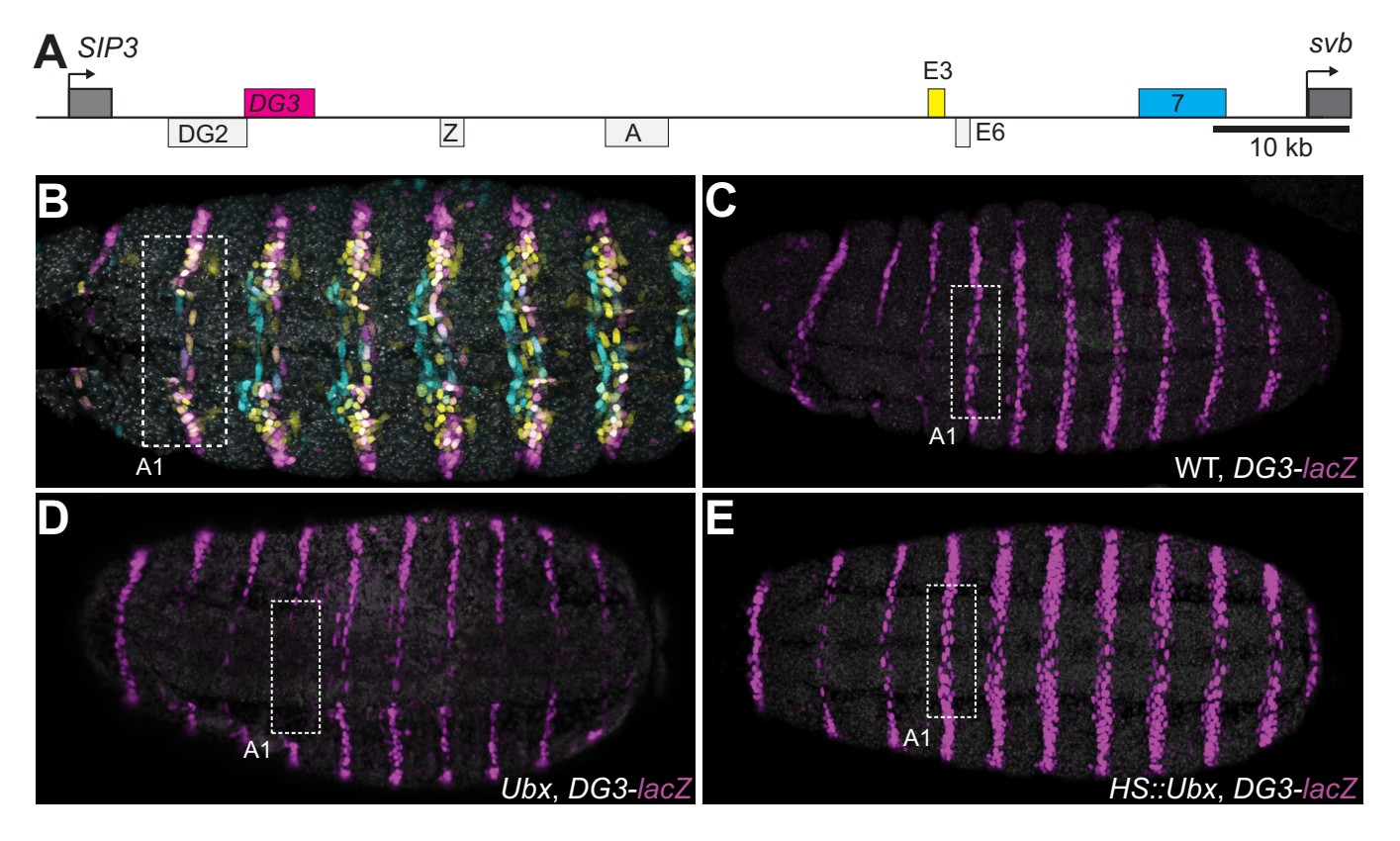

**Figure 1.** Ubx drives the expression of the *DG3 shavenbaby* enhancer along the ventral abdominal segments. (**A**) The *cis*-regulatory region of the *shavenbaby* (*svb*) gene contains three enhancers expressing stripes on the ventral side of the abdominal segments: *DG3*, *E3* and *7*. (**B**) The expression patterns of the three enhancers are partially overlapping. The color scheme corresponds to (**A**), where *DG3* is magenta, *E3* is yellow and *7* is cyan. (**C**) Expression pattern of a reporter construct with the *DG3* enhancer driving LacZ expression in an embryo with wild-type Ubx expression, as visualized using immunofluorescence staining. (**D**) *DG3* reporter in Ubx null mutant shows no expression in A1 and significantly weakened expression in the other abdominal segments. (**E**) Overexpression of Ubx driven through a heat shock promoter induces overexpression of *DG3* reporter in all abdominal segments and ectopic expression on the ventral surface of the thoracic segments T2 and T3.

DOI: https://doi.org/10.7554/eLife.45325.002

The following figure supplement is available for figure 1:

**Figure supplement 1.** Ubx enrichment around the three ventral *svb* enhancers.

DOI: https://doi.org/10.7554/eLife.45325.003

weak with wild-type Ubx expression and was primarily seen on the sides. In the absence of Ubx, *DG3* reporter expression was almost completely lost on the ventral side of A1 and reduced between A2-A7 (*Figure 1D*), consistent with the responses of *E3*, *7* and the full *svb* locus (*Crocker et al., 2015*). The incomplete loss of expression in A2-A7 suggests that additional factors influence the expression of ventral trichomes in those segments. Ubiquitous expression of Ubx increased the expression levels in A1-A7, in addition to generating ectopic expressions on the ventral side of the thoracic segments T2-T3 and A8 (*Figure 1E*). In summary, we showed that Ubx is necessary for *DG3* expression in the ventral region of the abdominal segments (completely in A1 and partially in A2-7) and can induce ectopic expressions when overexpressed. These data are consistent with our previous observation of the localization of *DG3*-driven transcription sites within Ubx microenvironments (*Tsai et al., 2017*).

## Deletion of a region including *DG3* enhancer causes defects in ventral trichome formation specifically at elevated temperatures

Given the clear ventral stripes that *DG3* generated in the abdominal segments, we next explored the phenotypic impact of its activity in driving trichome formation. It had been previously shown that deleting a region in the *svb* locus containing *DG3* (*Df(X)svb*[108]) led to reduced phenotypic robustness of *svb* under non-optimal temperatures, with reduced numbers of trichomes produced (*Frankel et al., 2010*). This *svb DG3*-deletion allele encompasses the enhancers *DG2*, *DG3* and *Z* (*Figure 2A*)—of which only *DG3* is a ventral enhancer.

In the A1 and A2 segments at 25°C, deletion of the *DG3* enhancer did not result in a clear change in ventral trichome formation in the abdominal segments compared to the wild-type (*Figure 2B and C*), perhaps due to the redundancy provided by overlapping expression patterns from other *svb* enhancers. However, the T1 trichomes were missing in larvae homozygous for the deletion (*Df(X) svb*[108]) allele (*Figure 2D and E*), which we subsequently used as a homozygous marker to select for larvae homozygous for the deletion allele when crossing *Df(X)svb*[108] flies to other lines (See 'Cuticle preparations and trichome counting' in Materials and Methods). Also, we observed defects in trichome formation in the dorsal edges of the stripe pattern, which are exclusively covered by *DG3* (*Figure 2D and E*, the black brackets at A1 and A2). This is consistent with a lack of redundancy in enhancer usage in these areas (*Figure 2F*, white dotted circles). The trichome number in regions covered by the overlapping expression of the *E3*, *7* and *DG3* enhancers in the A1 segment did not significantly reduce at 25°C upon the deletion of *DG3* (*Figure 2B,C and I*). However, larvae homozygous for the *Df(X)svb*[108] allele developed at 32°C produced fewer trichomes compared to wild-type larvae (*Figure 2G,H and I*). We also observed similar effects in the A2 segment (*Figure 2—figure supplement 1*). These results are similar to those shown with quartenary A5 trichomes (*Frankel et al., 2010*). However, the mechanisms behind this loss of phenotypic robustness under heat-induced stress are yet to be understood in detail.

## Transcription sites from the *DG3*-deletion allele have weaker Ubx microenvironment and lower transcriptional output

To address molecular sources that may lead to the reduced number of ventral trichomes we observed for the *Df(X)svb*[108] deletion allele, we imaged Ubx distributions and the transcriptional output of the *svb* gene in fixed *Drosophila melanogaster* embryos using high-resolution confocal microscopy. We reasoned that the defect could be due to changes in the transcription factor concentration around the enhancers (input) and/or the transcriptional output of the gene (output). As Ubx is specifically needed to drive *DG3* and *svb* expression on the ventral surface of the A1 segment, we used it as a metric for the transcription factor distributions around *svb* transcription sites. The samples were stained with immunofluorescence (IF) for Ubx and RNA fluorescence *in situ* hybridization (FISH) for *svb* transcription sites as previously described (*Tsai et al., 2017*). We imaged both embryos containing the wild-type *svb* allele or the *Df(X)svb*[108] allele, raised at either 25°C or 32°C. For all imaging experiments involving the *svb* allele with the deletion, we selected only homozygous embryos for imaging (See 'Imaging fixed embryos' in Materials and Methods). Because *DG3* expression in the A1 segment showed a clear link to changes in Ubx level, we focused most of our subsequent imaging quantifications in this segment.

To gauge the Ubx concentration around a transcription site, we counted the averaged intensity in the Ubx IF channel within a circle four pixels in diameter (170 nm, roughly the resolution limit of AiryScan) centered on the transcription site (*Figure 3A and B*, see 'Analysis of microenvironment and *svb* transcription intensity' in materials and methods). In nuclei from the A1 segment, Ubx intensities around *svb* transcription sites with the wild-type allele did not significantly change between 25°C and 32°C (*Figure 3C*, bottom right panel). Measuring Ubx intensity from random locations within nuclei of wild-type embryos expressing *svb* at 25°C using the same method showed that Ubx concentrations around *svb* transcription sites were in general higher than the nuclear average (*Figure 3—figure supplement 1*), consistent with our previous findings (*Tsai et al., 2017*). Transcription sites in embryos with the *DG3*-deletion (*Df(X)svb*[108]) allele had a local Ubx concentration that is slightly lower than wild-type at 25°C (*Figure 3C*, bottom right panel). However, there was a clear decrease in Ubx intensity compared to the wild-type when we subjected the *DG3*-deletion embryos to heat-stress (*Figure 3B*, right panel, and 3C, bottom right panel). To measure the transcriptional

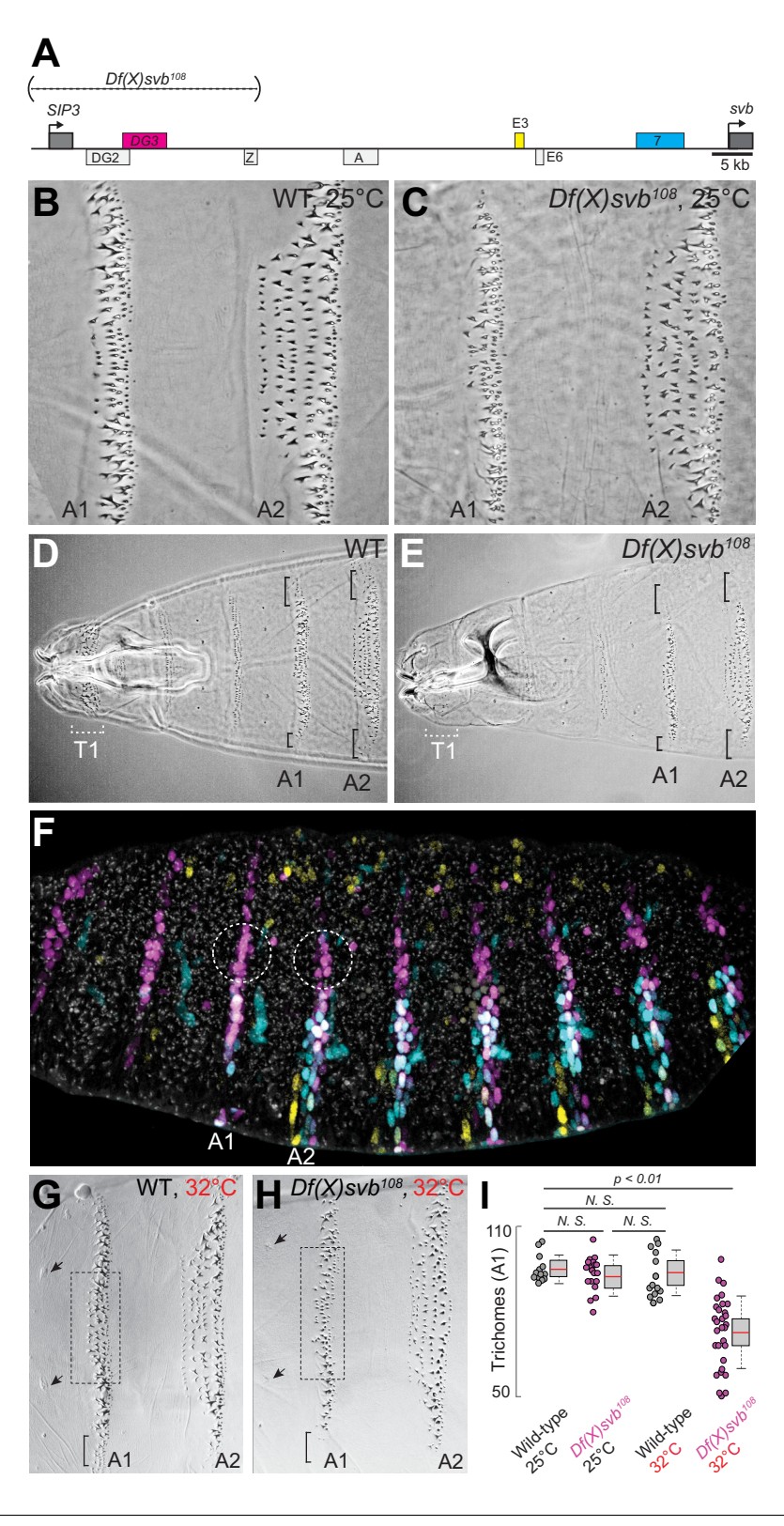

**Figure 2.** Deletion of a region from the *svb* locus containing *DG3* reduces ventral trichome numbers under heat-induced stress. (A) The *Df(X)svb^108* allele contains a deletion in the *cis*-regulatory of svb spanning three enhancers: *DG2*, *DG3* and *Z*. Of those, only *DG3* expresses on the ventral side. (B) Wild-type phenotype of trichomes along the A1 and A2 segments at 25°C. (C) At 25°C, the *Df(X)svb^108* deletion allele did not show a mutant phenotype along the A1 and A2 segments. (D) Zoomed out shot of the anterior region of a cuticle preparation of a wild-type (*w^1118*) larva. (E) Zoomed out shot of

*Figure 2 continued on next page*

*Figure 2 continued*

the anterior region of a cuticle preparation of a larva carrying *Df(X)svb*[108]. The lack of trichomes along the T1 segment is a recessive marker used in subsequent experiments to select for embryos/larvae carrying this deletion allele. Even at 25°C, where the overall trichome numbers in A1 and A2 for the *Df(X)svb*[108] deletion mutant is indistinguishable from wild-type *svb*, the trichomes at the side of the ventral stripe for A1 and A2 were lost, as marked by the black brackets. (F) Within the overall expression pattern of *svb* enhancers, *DG3* provides exclusive coverage in the circled regions in segments A1 and A2. These regions correspond to the black brackets in panels D and E. (G) Wild-type phenotype of trichomes along the A1 and A2 segments at 32°C. (H) At 32°C, the *Df(X)svb*[108] deletion allele showed a mutant phenotype. (I) A1 trichomes in the dashed boxes bounded by the two sensory cells, as indicated by the arrows, as shown in panels G and H, were counted. Deficiencies of the deletion allele only become clear when the animal is subjected to elevated temperature at 32°C, showing reduced trichome numbers in the A1 segment. The number of larvae counted was: 13 for wild-type at 25°C, 19 for *Df(X)svb*[108] at 25°C, 14 for wild-type at 32°C and 30 for *Df(X)svb*[108] at 32°C. Two-tailed t-test was applied for each individual comparison. In box plots, center line is the mean, upper and lower limits are standard deviation and whiskers show 95% confidence intervals.
DOI: https://doi.org/10.7554/eLife.45325.004

The following figure supplement is available for figure 2:

**Figure supplement 1.** Loss of trichomes in the A2 segment.
DOI: https://doi.org/10.7554/eLife.45325.005

output of *svb*, we adopted the same approach, but quantified the intensity in the *svb* RNA FISH channel (*Figure 3C*, upper left panel). Interestingly, we detected clear decreases in transcriptional output when the embryos are heat-stressed at 32°C, even with the wild-type allele. The *Df(X)svb*[108] allele at 25°C showed reduced levels of transcriptional output slightly lower than the wild-type under heat-shock. At 32°C, the transcriptional output further decreased in the mutant. When the various conditions were plotted by their *svb* transcriptional output and Ubx intensity (*Figure 3C*, center panel), they displayed a weak positive correlation with Ubx concentration (R-square = 0.4331). In sum, stress conditions reduced the transcriptional output of enhancers and the correlation to Ubx concentrations was positive but weak.

## *Df(X)svb*[108] deficiencies are rescued upon insertion of the full *svb cis*-regulatory region in a different chromosome

Having observed in the past that transcriptional microenvironments can be shared between related *svb* enhancers on different chromosomes (*Tsai et al., 2017*), we wondered whether this phenomenon could enhance transcriptional output and thus buffer against adverse environmental conditions. Therefore, we tested the capacity of a DNA sequence containing the full *svb cis*-regulatory region to rescue the described molecular and developmental defects of the *Df(X)svb*[108] allele (*Figure 4A*). For this purpose, we used a transgenic fly line, where a bacterial artificial chromosome (BAC) carrying the complete *cis*-regulatory region of *svb* (*Preger-Ben Noon et al., 2018*) was integrated into chromosome 2. To exclude *svb* mRNA from effecting the rescue, this *svbBAC* construct drives a *dsRed* reporter gene instead of another copy of *svb*. We confirmed that DsRed protein expression driven by this regulatory sequence recapitulated the *svb* expression patterns (*Figure 4B*) in *D. melanogaster* embryos and was responsive to Ubx—the lack of Ubx led to a decrease of expression in the A1 segment (*Figure 4C*).

To test the rescue, embryos or larvae with a *svbBAC-dsRed* crossed into them were incubated at 32°C. We observed that many active *dsRed* transcription sites were close to *svb* transcription sites in nuclei expressing both *svb* and *dsRed* in embryos from crosses between *svbBAC-dsRed* and wild-type (*w*[1118]) flies (*Figure 4D*). Similar observations were previously seen for the endogenous *svb* locus with itself and with the other two ventral enhancers (*E3* and *7*) (*Tsai et al., 2017*). This observation was also true for embryos from crosses between *svbBAC-dsRed* and *Df(X)svb*[108] flies, suggesting that the co-localization of related regulatory regions could occur under stressed conditions. This effect was not observed for the unrelated regulatory region of *diachete* driving the expression of *gfp*, which was inserted on a BAC in the same chromosomal location as *svbBAC*.

We observed that the introduction of the *svb* regulatory region was able to rescue both molecular and functional defects observed from the loss of the region containing *DG3*. Both *svb* transcriptional output (*Figure 4E*) and local Ubx concentration around *svb* transcription sites (*Figure 4F and G*) were restored to wild-type levels, but only when they co-localized with an active *svbBAC-dsRed* transcription site in the same nucleus (within 360 nm from each other, see 'Analysis of distances between transcription spots' in materials and methods for the definition of colocalization). The pairs of

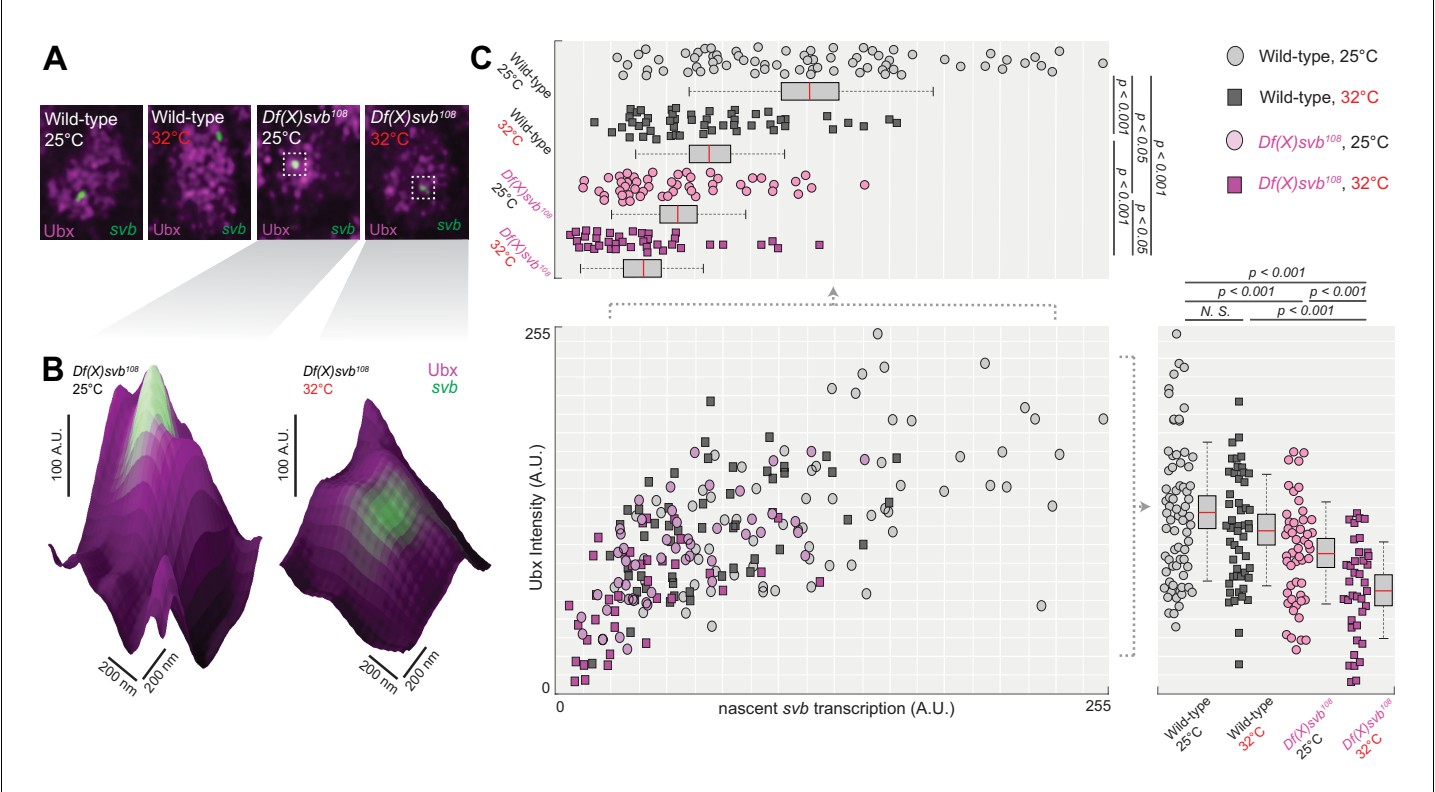

**Figure 3.** Deletion of the *cis*-region of *svb* containing *DG3* led to defects in the Ubx microenvironment and *svb* transcriptional output. (A) Panels showing a nucleus from embryos with either the wild-type (*w1118*) or *Df(X)svb108* deletion *svb* allele at either at normal (25°C) or elevated temperature (32°C), imaged using confocal fluorescence microscopy. Ubx (shown in magenta) is stained using immunofluorescence (IF) and the *svb* transcription sites (shown in green) are stained using fluorescence *in situ* hybridization (FISH). (B) Zoomed-in panes centered on *svb* transcription sites, with the height of the surface plots representing the Ubx intensity. (C) Correlation between Ubx and *svb* transcription intensities at different conditions. The number of transcription sites quantified was: 71 for wild-type at 25°C, 51 for wild-type at 32°C, 50 for *Df(X)svb108* at 25°C and 38 for *Df(X)svb108* at 32°C. Bottom right panel: Integrating the Ubx intensity surrounding transcription sites shows a small defect in the Ubx concentration around the deletion allele. The drop in Ubx increased at elevated temperature. Upper left panel: The integrated intensity of *svb* transcriptional output shows that there is a drop in transcriptional output for the deletion allele compared to the wild-type at both 25°C and 32°C. Even the wild-type showed reduced transcriptional output at elevated temperature (32°C). We analyzed four embryos for each genotype/temperature combination. Two-tailed t-test was applied for each individual comparison. In box plots, center line is mean, upper and lower limits are standard deviation and whiskers show 95% confidence intervals.
DOI: https://doi.org/10.7554/eLife.45325.006

The following figure supplement is available for figure 3:

**Figure supplement 1.** Ubx signal intensities in nuclei outside of transcription sites are consistent.
DOI: https://doi.org/10.7554/eLife.45325.007

transcription sites under and above this threshold clustered in two groups distinguishable also by Ubx and *svb* transcription levels (**Figure 4H**). Wild-type (*w1118*) x *svbBAC-dsRed* embryos and larvae were similar to wild-type in both trichome number and Ubx levels around *svb* transcription sites (**Figure 4—figure supplement 1**).

The phenotype, ventral trichome formation on the A1 segment (**Figure 5A–C**), which is reduced with the *DG3*-deletion allele, was partially rescued by the introduction of *svbBAC* (**Figure 5D**). The loss of the outer edge trichomes in A1 (in the black brackets in **Figure 5A–C**, where only *DG3* provides coverage **Figure 2F**) with the *DG3*-deletion allele was not rescued with *svbBAC*. Additionally, introducing only the *DG3* enhancer as opposed to *svbBAC* did not rescue trichome formation under heat-stress (**Figure 5D**).

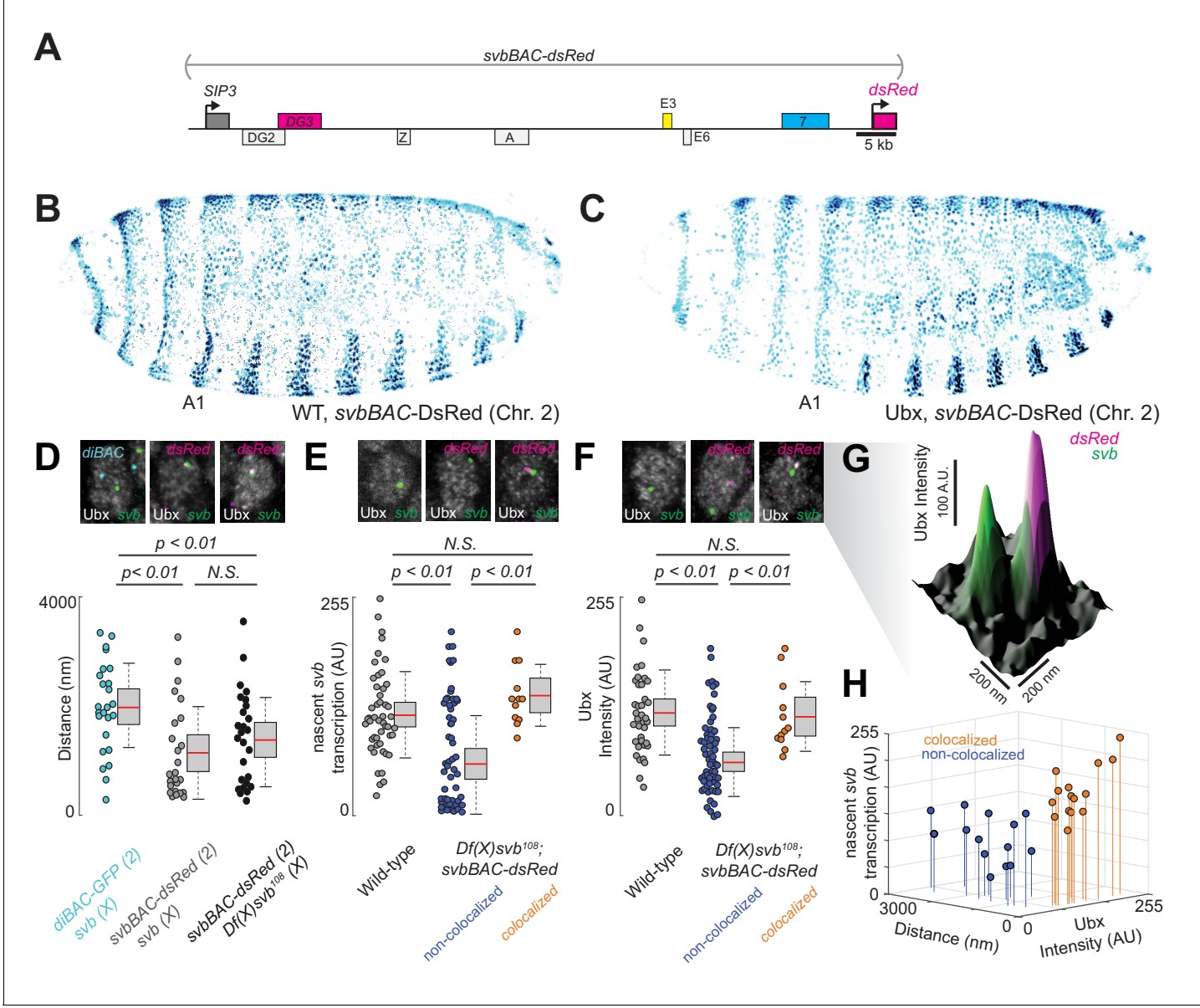

**Figure 4.** Introduction of the *cis*-regulatory region of *svb* on another chromosome rescues the microenvironment deficiencies of the *Df(X)svb*[108] deletion mutant. (**A**) The *svbBAC* construct encompasses the entire *cis*-regulatory region of *svb*, driving the expression of *dsRed*. (**B and C**) The *svbBAC* driving the expression of DsRed inserted into the second chromosome drives a similar expression pattern as the wild-type *svb* locus and responds similarly to Ubx. (**D**) In nuclei having both *svb* (for both the wild-type and the *Df(X)svb*[108] allele) and *svbBAC-dsRed* transcription sites at 32°C, the distances between them areon average closer than between that of *svb* and a reporter construct of an unrelated gene, *diachete* (*diBAC-gfp*, at 25°C), inserted into the same location as *svbBAC* on the second chromosome. The pairs of distance quantified were: 25 between *diBAC-gfp* and wild-type *svb*, 25 between *svbBAC-dsRed* and wild-type *svb* and 26 between *svbBAC-dsRed* and *Df(X)svb*[108]. (**E**) The *svb* FISH intensity (representing transcriptional output) in *Df(X)svb*[108] x *svbBAC-dsRed* embryos at 32°C recovered to wild-type levels when the *svb* transcription site is close to a *dsRed* transcription site (colocalized). FISH intensity of *svb* in the same embryos in nuclei without a *dsRed* transcription site or where *svb* and *dsRed* transcription sites were not near each other (non-colocalized) did not recover. The center image panel was stained for *dsRed* but the particular nucleus does not show *dsRed* signal. The number of transcription sites quantified was: 49 for wild-type, 53 for *Df(X)svb*[108] not near a *dsRed* transcription site (including cells where there were no *dsRed* transcription sites) and 12 for *Df(X)svb*[108] near a *dsRed* transcription site. (**F**) At 32°C, Ubx concentration around *svb* transcription sites recovered to wild-type levels in nuclei containing colocalized *svb* and *dsRed* transcription sites (colocalized) in *Df(X)svb*[108] x *svbBAC-dsRed* embryos. Ubx levels around transcription sites of *svb* in the same embryos in nuclei without a *dsRed* transcription site or where *svb* and *dsRed* transcription sites were not near each other (non-colocalized) did not recover. The number of transcription sites quantified was: 38 for wild-type, 60 for *Df(X)svb*[108] not near a *dsRed* transcription site (including cells where there were no *dsRed* transcription sites) and 12 for *Df(X)svb*[108] near a *dsRed* transcription site. Two-tailed t-test was applied for each individual comparison. In box plots, center line is mean, upper and lower limits are standard deviation and whiskers show 95% confidence intervals. (**G**) A surface plot (the height representing Ubx intensity) showing two *svb* transcription

*Figure 4 continued on next page*

*Figure 4 continued*

sites in a nucleus, with the one on the right overlapping with a *svbBAC-dsRed* transcription site and showing higher Ubx concentration. (**H**) At 32°C, in nuclei having both *svb* (from the *Df(X)svb*[108] allele) and *svbBAC-dsRed* transcription sites, the distances between them are plotted against *svb* FISH intensity and Ubx intensity. There are two clusters separated by a threshold of 360 nm in distance. Co-localized pairs below this distance threshold present higher intensities for both Ubx intensity and nascent *svb* transcription. The pairs quantified were 15 for colocalized and 14 for non-colocalized between *svbBAC-dsRed* and *Df(X)svb*[108]. We analyzed four embryos for *diBAC-gfp* x *w*[1118], four embryos for *svbBAC-dsRed* x *Df(X)svb*[108] and for five embryos for *svbBAC-dsRed* x *w*[1118].

DOI: https://doi.org/10.7554/eLife.45325.008

The following figure supplement is available for figure 4:

**Figure supplement 1.** Introduction of *svbBAC-dsRed* to wild-type (*w*[1118]) does not change Ubx microenvironment and phenotype.
DOI: https://doi.org/10.7554/eLife.45325.009

## Discussion

Transcriptional regulation is a complex and dynamic process which requires coordinated interactions between transcription factors and chromatin. Given the transient nature of these interactions, using multiple binding sites to ensure efficient and consistent transcriptional regulation under different environmental conditions appears to be a preferred strategy among many developmental enhancers (*Frankel, 2012*; *Perry et al., 2010*). Genes such as *shavenbaby* add another layer of redundancy on top of this through long *cis*-regulatory regions containing multiple enhancers whose expression patterns overlap. Previous works have shown that this redundancy ensures proper phenotype development when systems are subjected to stress (*Crocker et al., 2015*; *Frankel et al., 2010*; *Osterwalder et al., 2018*). However, the mechanism underlying this phenotypic robustness was not clear.

In this work, we took advantage of the high-resolution imaging and analysis techniques we had developed to observe transcriptional microenvironments around transcription sites (*Tsai et al., 2017*) and investigated how the *DG3* enhancer contributes to the phenotypic robustness of the *svb* locus at the molecular level. Deletion of the *DG3* enhancer from *svb* did not lead to clear defects in ventral trichome formation unless the embryos were subjected to heat-induced stress, as shown

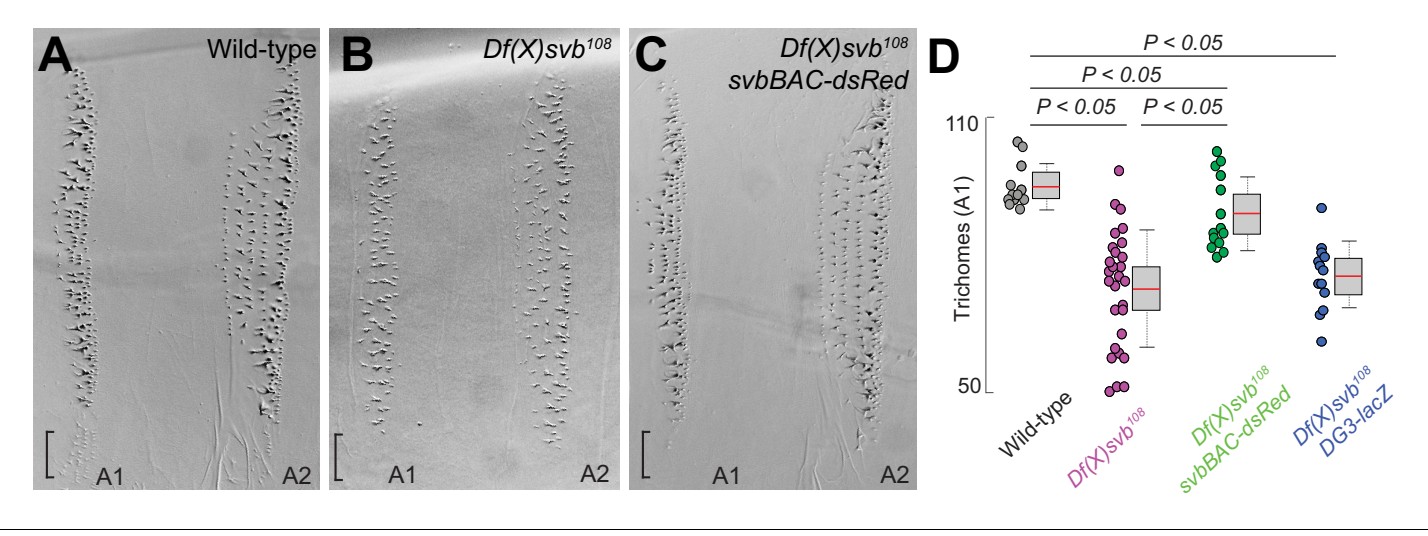

**Figure 5.** The complete *cis*-regulatory region of *svb* rescues trichome number. (**A–C**) Cuticle preparations from larvae developed at 32°C with wild-type *svb*, *Df(X)svb*[108] and *Df(X)svb*[108] x *svbBAC-dsRed*. The bracket at the edge of the A1 stripe marks a region where trichome growth is exclusively covered by *DG3*, which disappeared with the deletion of *DG3* and did not recover with the introduction of *svbBAC-dsRed*. (**D**) The trichome number in larvae developed at 32°C with *Df(X)svb*[108] partially recovered to wild-type levels with the introduction of *svbBAC-dsRed*. The number of larvae counted was: 12 for wild-type *svb*, 28 for *Df(X)svb*[108], 14 for *Df(X)svb*[108] x *svbBAC-dsRed* and 13 for *Df(X)svb*[108] x *DG3-lacZ*. Two-tailed t-test was applied for each individual comparison. In box plots, center line is mean, upper and lower limits are standard deviation and whiskers show 95% confidence intervals.
DOI: https://doi.org/10.7554/eLife.45325.010

here and previously with the deletion of 'shadow enhancers' (*Hong et al., 2008*) for lateral *svb* expression (*Frankel et al., 2010*). Nevertheless, we observed that the *DG3*-deletion allele showed reduced transcriptional output even at normal temperature. Wild-type embryos did not show phenotypic defects under either normal or stressed conditions, but heat-induced stress led to slightly lower transcriptional output from the wild-type *svb* allele (*Figure 6A and B*). However, the mutant *svb* locus, starting with lower transcriptional output even under ideal conditions (*Figure 6C*), drops below a threshold and the system fails (*Figure 6D*). We observed that Ubx concentration and *svb* transcriptional output in the A1 segment initially have a weak positive correlation that quickly dissipated at higher Ubx concentrations and *svb* outputs. While this data must be interpreted with some caution, as multiple stages of the transcription cycle are included (initiation, elongation or termination), Ubx concentration increases after a certain point were not clearly coupled to increases in *svb* transcriptional output. Additional regulatory mechanisms could be at play beyond transcription factor retention that determines the final transcriptional output of the locus. Determining the complete response function would be complicated, due to reasons such as the enhancers each having clusters of factor binding sites and the overlap of expression patterns from related enhancers. As *svb* also accepts inputs from many additional transcription factors (*Stern and Orgogozo, 2008*), especially in the other body segments, the total response of the system would also depend on many factors not observed in this study.

We previously observed that transcription sites of reporter genes driven by minimal *svb* enhancers tended to colocalize with the endogenous *svb* locus when it is transcriptionally active (*Tsai et al., 2017*). This is true also for entire *cis*-regulatory regions, as we observed that the *svb* locus did the same with *svbBAC*, which implies that they potentially share a common microenvironment. Homologous regions were shown to pair over long distances, between homologous chromosomal arms (*Lim et al., 2018*), translocated domains and even different chromosomes (*Gemkow et al., 1998*; *Johnston and Desplan, 2014*; *Peifer and Bender, 1986*). Our observations could be related to them and share similar mechanisms. However, our constructs may not contain sites for structural elements such as insulators, which have been described as important supporters of *trans*-interactions (*Postika et al., 2018*). On the other hand, our observations are in line with transcription-dependent associations of interchromosomal interactions (*Branco and Pombo, 2006*; *Joyce et al., 2016*; *Lomvardas et al., 2006*; *Maass et al., 2018*; *Monahan et al., 2019*). It is possible that such long range interactions are driven, or reinforced, through shared microenvironments.

We were able to partially rescue the *DG3*-deletion *svb* allele with *svbBAC*, which contains the *cis*-regulatory region of *svb* but not the *svb* gene itself. High-resolution imaging showed that colocalizing with a *svbBAC* increases the local Ubx concentration and transcriptional output of the *DG3*-deletion allele (*Figure 6E*). This supports a mechanism where transcriptional microenvironments sequestered around large and related *cis*-regulatory regions in physical proximity can work *in trans* to increase transcriptional output of other genes, even on different chromosomes. The introduction of a shorter reporter construct containing the *DG3* enhancer alone did not rescue trichome expression perhaps because it is not able to effectively pair with the *svb* locus. It is possible that structural elements, such as insulator proteins (*Lim et al., 2018*) or other topologically associated elements (*Furlong and Levine, 2018*) could overcome this by increasing pairing efficiency. This is consistent with recent findings for long-range interactions that are dependent on specific topologically associating domains (TADs), where pairing is necessary but not sufficient for transvection (*Viets et al., 2018*). Interestingly, embryos with the *DG3* deletion allele and *svbBAC* could not produce trichomes at the dorsal edges of the ventral trichome patches, where *DG3* provides exclusive coverage. As the rescue only occurred on regions where other ventral *svb* enhancers provided overlapping coverage, it likely is the result of compensation from the additional ventral enhancers (*E3* and *7*) at the *svb* locus instead of the *DG3* enhancer on the *svbBAC* driving *svb* expression *in trans*. Interactions *in trans* may serve an auxiliary role through influencing the properties of the local transcriptional environments.

Summarizing our observations, we hypothesize that the relationship between *svb* transcriptional output and the number of cells fated to become trichomes is sigmoidal (*Figure 6*, bottom panel). Below a certain threshold, the number of trichomes would drop with decreasing *svb* mRNA; however, any additional transcriptional output above this threshold (the green box in the figure) would not lead to significant changes in trichome production and would appear to be wild-type in phenotype. The *cis*-regulatory region of wild-type *svb* under ideal conditions likely already saturates the

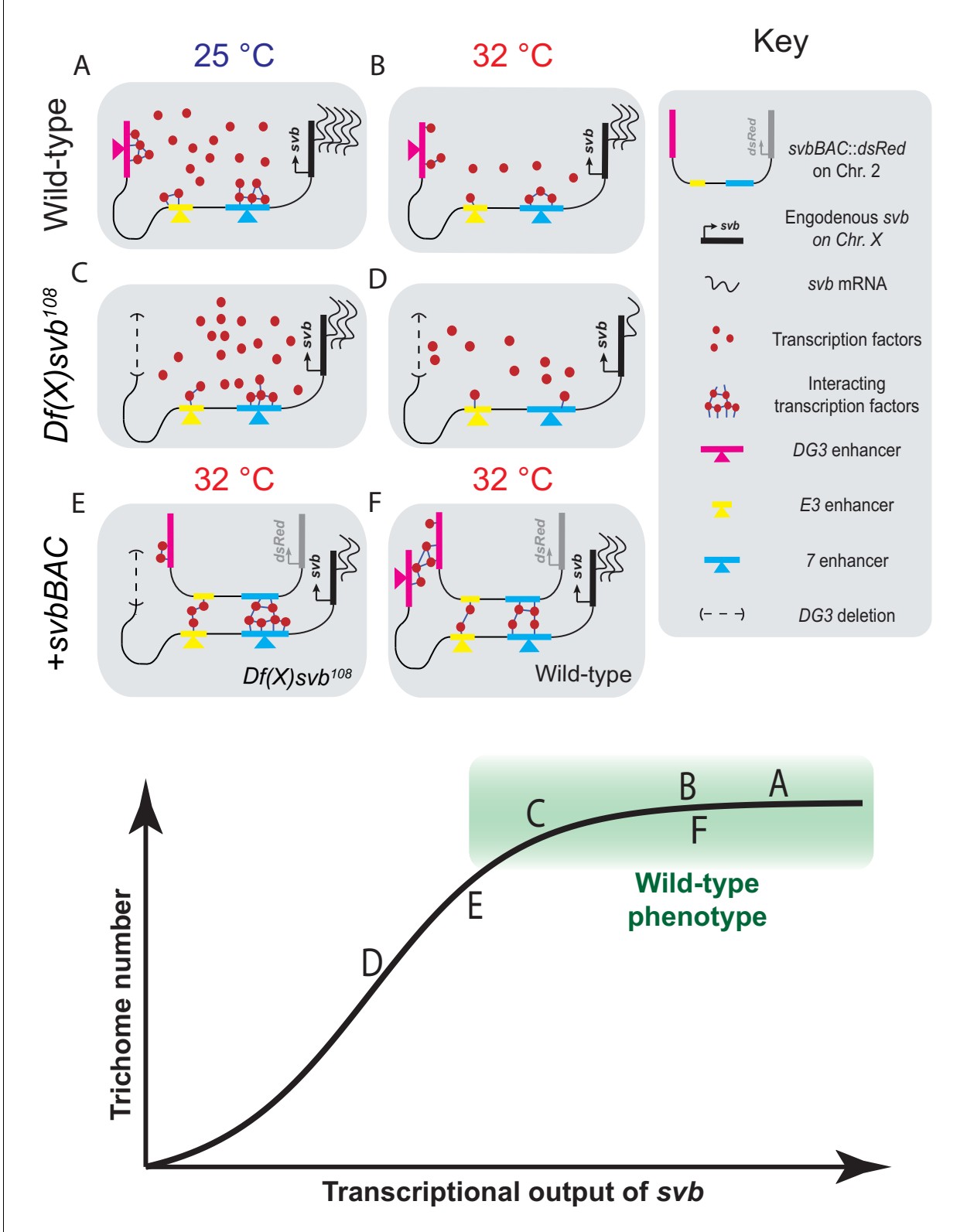

**Figure 6.** Summary of model including a sigmoidal relationship between *svb* transcriptional output and the number of cells fated to become trichomes. (A–F) Schematic representations of the various genotypes, temperatures and rescue conditions as tested in this work. Bottom panel: Schematic of the proposed sigmoidal relationship between *svb* transcriptional output and the number of cells fated to become trichomes. Note that A and C are at 25°C

*Figure 6 continued on next page*

*Figure 6 continued*

and the rest are at 32°C. The different conditions we have studied are represented by letters A-F along the sigmoidal curve. The part of the curve corresponding to wild-type phenotype is shaded in green.

DOI: https://doi.org/10.7554/eLife.45325.011

system, as evidenced by the lack of change in Ubx concentration and trichome numbers even with the addition of *svbBAC* (*Figure 6F*). Although operating under saturation renders this system relatively insensitive to changes in *svb* transcription, the risk of developing defective phenotypes when conditions are no longer ideal likely selected for this strategy to buffer against stresses. Overall, the system remains phenotypically robust and develops the same number of trichomes despite fluctuations in transcriptional outputs. In the future, it would be important to understand mechanistically how the phenotype can tolerate significant drops in transcriptional output before defects appear. It requires the direct observation of intermediate steps between *svb* transcription and phenotype production to understand how this buffering is achieved. Furthermore, our current technique only allows us to probe environments around active transcription sites without knowing which phase of transcription (initiation, elongation or termination) it is in and how the gene loci positioned themselves into these locations. Future works to visualize and track genes, regardless of their transcriptional state, in fixed and living embryos would answer key questions on how they find, form and interact with transcriptional microenvironments.

We previously proposed that transcriptional microenvironments form across multiple enhancers through scaffolding interactions to ensure efficient transcription from developmental enhancers. By investigating the mechanisms of how correct transcriptional regulation is maintained under stress using a *DG3*-deletion allele of *svb*, we have shown that transcriptional microenvironments could span multiple enhancers that share similar transcription factor binding sites. These microenvironments of transcription factors could form the protein core of transcription factor 'hubs' that have been proposed to form through phase-separation mediated through protein-protein interactions between disordered domains (*Cisse et al., 2013*; *Furlong and Levine, 2018*; *Ghavi-Helm et al., 2014*; *Mir et al., 2017*). Thus, they add another layer of redundancy on top of using multiple enhancers with overlapping expression patterns in a *cis*-regulatory region to ensure a sufficient margin to buffer against the negative effects of environmental stresses (*Frankel et al., 2010*; *Perry et al., 2010*). This extra margin of safety preserves phenotypical development even when environmental conditions are not ideal. Integrating multiple noisy and low-affinity elements into a coherent and synergistic network would also reduce the variance stemming from the transient and stochastic transcription factor binding dynamics observed in eukaryotic cells (*Cisse et al., 2013*; *Ghavi-Helm et al., 2014*; *Mir et al., 2017*; *Tsai et al., 2017*). In sum, specialized transcriptional microenvironments could be a critical element to ensure that gene expression occurs specifically and consistently in every embryo. Given that shadow enhancers are widespread features of gene regulatory networks (*Cannavò et al., 2016*; *Osterwalder et al., 2018*), it is likely that high local concentrations of transcription factors are a widespread feature that provides an effective regulatory buffer to prevent deleterious phenotypic consequences to genetic and environmental perturbations.

## Materials and methods

### Fly strains

All fly strains used have been previously described: *DG3-lacZ* (*Tsai et al., 2017*); *ubx1* (*Crocker et al., 2015*); *HS::ubx-1*: (*Crocker et al., 2015*); *Df(X)svb[108]* (*Frankel et al., 2010*); *svbBAC-dsRed* (*Preger-Ben Noon et al., 2018*); *diBAC-gfp* is CH322-35A16 EGFP tagged in VK37, covering D (*Venken et al., 2009*). Unless otherwise noted, they are generated from *w[1118]* stock, which is referred to as wild-type.

### Preparing *Drosophila* embryos for staining and cuticle preps

*D. melanogaster* strains were maintained under standard laboratory conditions, reared at 25 °C, unless otherwise specified. For heat-shock experiments, these conditions were followed: for staining with fluorescent antibodies, flies were allowed to lay eggs on apple-juice agar plates for 5 hr at 25 °

C and then kept in an incubator at 32 °C for 7 hr before fixation; for cuticle preps, dechorionated embryos were kept at 32 °C until they emerged as larvae. *Df(X)svb*[108] embryos/larvae with *svbBAC-dsRed* are readily discernable by the loss of *svb* and trichomes in the T1 segment (see *Figure 2D & E*).

### Cuticle preparation and trichome counting

Larvae collected for cuticle preparations were mounted according to a published protocol (*Stern and Sucena, 2011*). A phase-contrast microscope was used to image the slides. In the case of crosses involving the *Df(X)svb*[108] allele, only larvae lacking trichomes in the T1 segment were imaged (*Figure 2E*). This is a homozygous marker for the deletion locus as larvae carrying any wild-type *svb* allele will produce trichomes in T1 (*Figure 2D*). Ventral trichomes in larval A1 or A2 segments were counted in Fiji/ImageJ by find using the find maximum function (*Schindelin et al., 2012*; *Schneider et al., 2012*).

### Immuno-fluorescence staining of transcription factors and *in situ* hybridization to mRNA

Standard protocols were used for embryo fixation and staining (*Crocker et al., 2015*; *Tsai et al., 2017*). Secondary antibodies labeled with Alexa Fluor dyes (1:500, Invitrogen) were used to detect primary antibodies. *In situ* hybridizations were performed using DIG, FITC or biotin labeled, anti-sense RNA-probes against a reporter construct RNA (*lacZ*, *dsRed*, *gfp*) or the first intron and second exon (16 kb) of *svb*. See *Supplementary file 1* for primer sequences. DIG-labeled RNA products were detected with a DIG antibody: Thermofisher, 700772 (1:100 dilution), biotin-labeled RNA products with a biotin antibody: Thermofisher, PA1-26792 (1:100) and FITC-labeled RNA products with a FITC antibody: Thermofisher, A889 (1:100). Ubx protein was detected using Developmental Studies Hybridoma Bank, FP3.38-C antibody at 1:20 dilution, DsRed protein using MBL anti-RFP PM005 antibody at 1:100, LacZ protein using Promega anti-ß-Gal antibody at 1:250 and GFP protein using Aves Labs chicken anti-GFP at 1:300.

### Imaging fixed embryos

Mounting of fixed *Drosophila* embryos was done in ProLong Gold + DAPI mounting media (Molecular Probes, Eugene, OR). Fixed embryos were imaged on a Zeiss LSM 880 confocal microscope with FastAiryscan (Carl Zeiss Microscopy, Jena, Germany). Excitation lasers with wavelengths of 405, 488, 561 and 633 nm were used as appropriate for the specific fluorescent dyes. For imaging in embryos carrying *Df(X)svb*[108], only embryos without *svb* mRNA expression in the T1 segment were imaged, following the same reason described in the section on 'Cuticle preparations and trichome counting'. Unless otherwise stated, all images were processed with Fiji/ImageJ (*Schindelin et al., 2012*; *Schneider et al., 2012*) and Matlab (MathWorks, Natick, MA, USA).

### Analysis of microenvironment and *svb* transcription intensity

Inside nuclei with *svb* transcription sites, the center of the transcription site was identified using the find maximum function of Fiji/ImageJ. A circle with a diameter of 4 pixels (170 nm, roughly the lateral resolution limit of AiryScan in 3D mode) region of interest (ROI) centered on the transcription site is then created. The integrated fluorescent intensity inside the ROI from the Ubx IF channel and the RNA FISH channel are then reported as the local Ubx concentration and the transcriptional output, respectively. The intensity presented in the figures is the per-pixel average intensity with the maximum readout of the sensor normalized to 255.

### Analysis of distances between transcription spots

Inside nuclei with *svb* and *dsRed/gfp* transcription sites, the centers of the transcription site were identified using the find maximum function of Fiji/ImageJ. The distance between the transcription sites were then computed using the coordinates of the transcription sites. Two sites are considered colocalized when they are within 360 nm of each other.

### Ubx ChIP profile

The ChIP profile for Ubx around the svb cis-regulatory region is from *Choo et al. (2011)*, using whole *Drosophila melanogaster* embryos between stages 10 and 12.

## Acknowledgements

The fly line containing *diBAC-gfp* (CH322-35A16 EGFP) was a gift from Schulze, Karen Lynn and Bellen, Hugo J. The *Df(X)svb*[108] flies were a gift from Stern DL and the *svb*BAC flies were a gift from Preger Ben-Noon E and Frankel N (*Preger-Ben Noon et al., 2018*). We thank Rafael Galupa, Nicolas Frankel and Ella Preger Ben-Noon for suggestions and discussions. We thank the entire Crocker lab for discussion and feedback. We would also like to thank the reviewers for their constructive input. Albert Tsai is a Damon Runyon Fellow of the Damon Runyon Cancer Research Foundation (DRG 2220–15). Albert Tsai, Mariana R P Alves and Justin Crocker are supported by the European Molecular Biological Laboratory (EMBL).

## Additional information

### Funding

| Funder | Grant reference number | Author |
|---|---|---|
| Damon Runyon Cancer Research Foundation | DRG 2220-15 | Albert Tsai |
| European Molecular Biology Organization | | Albert Tsai<br>Mariana R P Alves<br>Justin Crocker |

The funders had no role in study design, data collection and interpretation, or the decision to submit the work for publication.

### Author contributions

Albert Tsai, Conceptualization, Data curation, Formal analysis, Validation, Investigation, Visualization, Methodology, Writing—original draft, Writing—review and editing; Mariana RP Alves, Data curation, Formal analysis, Validation, Investigation, Visualization, Methodology, Writing—original draft, Writing—review and editing; Justin Crocker, Conceptualization, Data curation, Formal analysis, Supervision, Funding acquisition, Validation, Investigation, Visualization, Methodology, Writing—original draft, Project administration, Writing—review and editing

### Author ORCIDs

Albert Tsai ◎ https://orcid.org/0000-0002-1643-0780
Mariana RP Alves ◎ https://orcid.org/0000-0002-0796-2101
Justin Crocker ◎ https://orcid.org/0000-0002-5113-0476

### Decision letter and Author response

Decision letter https://doi.org/10.7554/eLife.45325.016
Author response https://doi.org/10.7554/eLife.45325.017

## Additional files

### Supplementary files

• Supplementary file 1. Primers for RNA-probe generation Sequences of primers for amplification of DNA to be used for generation of antisense RNA-probes. The targets - reporter construct RNA (*lacZ*, *dsRed*, *gfp*) and first intron and second exon (16 kb) of *svb* - are indicated in the left column. Sequences are indicated for forward or reverse primers of each pair. Reverse primers include a T7 sequence for transcription with T7 RNA polymerase.
DOI: https://doi.org/10.7554/eLife.45325.012

• Transparent reporting form
DOI: https://doi.org/10.7554/eLife.45325.013

**Data availability**

The original images (cuticle preparations and embryo images, organized into zip files) are available for download and are indexed at: https://www.embl.de/download/crocker/svb_enhancer_colocalization/index.html. Please note that the raw AiryScan images must be processed though the Zen software from Zeiss before they can be opened/analyzed using standard image processing softwares. These files are large, totaling up to approximately 180 GB in size. We can also send these files directly if a means of transfer (hard drives, etc.) is provided.

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
