## [Decision Letter]

Thank you for sending your article entitled "Multi-enhancer transcriptional hubs confer phenotypic robustness" for peer review at *eLife*. Your article is being evaluated by three peer reviewers, and the evaluation is being overseen by a Reviewing Editor and Jessica Tyler as the Senior Editor.

Given the list of essential revisions, including new experiments, the editors and reviewers invite you to respond within the next two weeks with an action plan and timetable for the completion of the additional work. We plan to share your responses with the reviewers and then issue a binding recommendation.

Essential points to address are the following:

1) The connection between enhancers, Ubx protein concentration, transcriptional output, and phenotype. Although different concentrations of Ubx are measured at the deleted vs. wild-type locus, an important point that the three reviewers agree upon is that gene expression levels and Ubx concentrations are not well correlated. There appear to be other unexplained factors that are influencing the expression of *svb* that do not correlate with the TF concentration. Thus, it is unclear whether the experiments described here support or reject the hypothesis laid out in the "multi-enhancer "hubs" improve robustness by increasing transcription factor retention near transcription sites."

The authors must consider further whether the global measurements shown in Figure 3 are the most effective measure of this proposed correlation, or whether a more fine-grained analysis might support the idea that boosted Ubx levels from complexed enhancers are driving transcription. As noted below, the authors might show scatter plots of Ubx concentration vs. *svb* transcription within each condition. A positive correlation would support their hypothesis.

The trichome phenotype is also not directly correlated to the transcriptional outputs and Ubx concentrations, and as mentioned in the Discussion, further layers of regulation are likely in play. This idea needs to be more fully fleshed out.

A finding related to "enhancer hubs boosting local concentration" is the finding that an ectopic BAC appears not to influence Ubx concentration in a wild-type *svb* background, only in a deletion background. Does this finding indicate that such hub formation is saturable? And if so, does this indicate that the wild-type *svb* locus forms a local hub with enhancers in the immediate vicinity, and trans-complementation is not a normal feature of *svb* function?

2) A second point raised concerned the exact cis regulatory regions in play. A minimal *DG3* enhancer drives gene expression in ventral abdominal stripes, but does not rescue Ubx concentrations from a trans setting. A larger deletion that includes *DG3* as well as additional Ubx binding regions (that are not sufficient for, but may be part of, ventral *DG3*-related activity) impacts transcription, trichome development, and robustness. A yet larger cis-regulatory domain on a BAC rescues some aspects of gene expression and Ubx concentration. The interpretation conflates *DG3* with the function of the deleted region; reviewers noted that a more careful interpretation would differentiate results from each of these different cis elements. For instance, the lack of trans-rescue by the *DG3* enhancer alone may be due to the inability of a short segment to transvect effectively. The interpretation should explicitly take into account known properties of transvecting regulatory loci in *Drosophila*.

3) Several aspects can be addressed by better justification and explanation of methods and data presentation, including

– Why sometimes either A1 or A2 trichomes are quantitatively assessed, depending on the figure;

– The use of one-tailed (vs. two-tailed) T-tests for statistical relevance;

– Recommendations for inclusion of both 25C and 32C phenotypes and data uniformly, and;

– Clarification of exact role of Ubx in T1-T3 regulation, as the conclusions drawn about *DG3* and Ubx roles are difficult to know based on the single images shown;

– The number of data points in Figures 3 and 4 are limited; is there a technical limitation to more extensive sampling of the transcriptional readouts and Ubx intensities?

4) In some cases, the actual experimental approach was unclear to the reviewers:

– there was some confusion about whether the *DG3* deletion mutants were homozygous, so there would be no wild-type copy of the gene in these embryos – is that indeed the case?

– Figure 4 shows *svb*, but not *dsRed* mRNA expression; is that correct? Imaging of *dsRed* only used to score a *svb* locus as "overlapped" vs. "not overlapped"?

– Choice of pixel size for ROI.

– Signal to noise for Ubx across the nucleus, as well as variation in average Ubx from sample to sample.

*Reviewer #1:*

This paper addresses a timely topic, namely how multiple enhancers may coordinate to regulate gene expression. This is being actively investigated in a number of systems using both case studies and genome wide methods, and the field is fueled by new capabilities in high resolution microscopy. Thus, interest in the topic is high. However, I have substantial concerns about how this work contributes to this topic.

The central hypothesis of this work is that increased levels of Ubx, provided by micro-environments created through coalescing multiple Ubx-responsive enhancers, increases the level of transcription of shavenbaby, *svb*; this increased level of transcription allows *svb* expression to be robust to high temperature variation. However, the evidence presented in this paper does not consistently support this hypothesis.

Let's first consider the set of experiments correlating the presence of the enhancer deletion (via a deficiency), the concentration of Ubx and the level of *svb* transcription.

– Deleting an *svb* enhancer (*DG3*) decreases resilience to high temperature stress, resulting in smaller numbers of trichomes in A1.

– In the A1 segment, *svb* enhancers localize to a microenvironment where Ubx concentrations are high compared to background.

– Deleting an *svb* enhancer decreases the concentration of Ubx in these microenvironments moderately at high temperature, but has no effect at low temperature.

– At high temperatures, transcription of *svb* decreases at both WT and deficiency alleles.

– At low temperatures, transcription of *svb* decreases under low temperature stress for the deficiency allele.

These results do not support the hypothesis because *svb* transcription is not consistently correlated with Ubx levels in the microenvironment.

A second concern is the interpretation of the rescue experiment, where a BAC containing the full *svb* locus is integrated onto another chromosome. This partially rescues the trichome phenotype of the deficiency allele. However, the Ubx levels in microenvironments were unchanged from WT. If the extra *svb* regulatory region associates with the endogenous *svb* transcription site, this should either result in above WT levels of Ubx, or there is some additional layer of regulation. This was not discussed.

*Reviewer #2:*

In this paper, Tsai et al. studied the general problem of how low affinity binding sites, which promote enhancer specificity, also permit robust activation. They focus on ventral "shadow" enhancers for shavenbaby/*svb*.

The authors start by focusing on the *DG3* enhancer, which was less studied before, and responds to Ubx in A1. Indeed, ChIP-seq data shows Ubx binding in a *DG3* element that drive reporter gene expression in trichome forming territories, and this reporter responds to perturbations of Ubx.

A deletion, *Df(x)svb^108^*, removes *DG3*, as well as several other Ubx-bound regions, and causes trichome formation defects only upon heat stress, and primarily in the region of *DG3* activity.

State-of-the-art imaging of *svb* transcription and Ubx protein distribution show a reduction of Ubx concentration in the vicinity of active *svb* loci, specifically in *Df(X)svb^108^*mutant cells under heat stress, and a reduction of transcriptional output upon heat stress, even in wild-type embryos.

Remarkably, exogenous *svb* cis-regulatory DNA could rescue the formation of a Ubx-rich "hub" containing the endogenous *svb* locus, and *svb* transcription, in *Df(X)svb^108^* mutants, but only when the two loci colocalized.

Taken together, the data presented is consistent with the notion that multiple enhancers form a Ubx-rich microenvironment necessary for robust *svb* expression and trichome formation. While the specifics are rather specialized, the problem of transcriptional hub formation and robust gene expression is a general problem of importance.

Nevertheless, there are still specific issues with the main conclusions based on the work presented, which seems somewhat preliminary.

According to Figure 2, *Df(X)svb^108^*removes more than *DG3*, including elements that clearly bind Ubx as per Figure 1—figure supplement 1, so it is not clear if the observed defects can be attributed to loss of *DG3*. Ideally, CRISPR/Cas9 should be used to remove Ubx-bound regions outside *DG3* (e.g. Ubx-bound DG2) and to remove *DG3* specifically, in order to support the conclusions that the observed effects of *Df(X)svb^108^* are due to a loss of *DG3*.

*DG3* remains active in the most dorso-lateral cells in Ubx mutant (Figure 1), in a domain that seems to be specifically marked by *DG3* reporters (Figure 1). This domain is portrayed as primarily affected by reduction of trichome formation in *Df(X)svb^108^* mutants, but then it is not clear that this defect is due to a lack of Ubx activity on *DG3* (both because *DG3-LacZ* seems to remain active and because the region removed by *Df(X)svb^108^*binds Ubx outside *DG3*).

In Figure 3, the discrepancy between robust accumulation of Ubx, except in *Df(X)svb^108^*embryo under stress, and the impact of heat stress on transcription even in wild-type embryos further indicates that changes in Ubx accumulation is only part of the explanation for both the *Df(X)svb^108^* phenotype and the response to heat stress.

The rescue presented in Figure 4 is a strong result but also difficult to interpret: for instance, it seems likely that the "hub" forms independently of Ubx, which presumably cannot accumulate on the *Df(X)svb^108^*chromosome, and then help place the *svb* locus in a Ubx-rich microenvironment. This is implied but not explicitly stated, or clearly demonstrated by the data.

For instance, Figure 5 adds to the notion that regulatory inputs other than *DG3* and Ubx govern the formation of the *svb* containing micro-environment and *svb* transcription. As acknowledged also in the Discussion, robustness of trichome formation also involves mechanisms downstream of *svb* transcription, making the link between Ubx hubs and robust trichome formation even more complex.

Taken together, a few of the criticisms above question whether the accumulation of Ubx, or the lack thereof, in microenvironments containing the *svb* locus explains the phenotypes described for *Df(X)svb^108^*. Rescue experiments more directly combining loss of *DG3* function with gain of Ubx function should help resolve these issues.

*Reviewer #3:*

In their previous work, Tsai, et al. observed that, during fruit fly embryogenesis, actively transcribing shavenbaby (*svb*) loci preferentially colocalized with regions of elevated concentration for the transcription factor Ubx, a known regulator of *svb* activity. Moreover, they observed that other Ubx target genes exhibited a tendency to colocalize to these same Ubx-enriched "microenvironments", even when located on different chromosomes. Based on these findings, they hypothesized that the spatial overlap between multiple distal enhancers and transcription factor microenvironments could result in increased transcriptional output and could be a mechanism for redundancy in the face of environmental stresses. In the present work, Tsai, et al. test this hypothesis directly by deleting the *DG3* enhancer from the *svb* cis-regulatory region and assessing the effects on microenvironment formation, *svb* transcription, and trichome number (a phenotype tied to *svb* expression) under normal (25C) and heat-stress (32C) conditions. This manuscript constitutes an intriguing extension of the findings presented in the authors' earlier work on Ubx microenvironments in the vicinity of *svb* loci. They find that the deletion mutant results in a significant depletion in Ubx concentration around the mutated *svb* locus at 32C, and that this depletion coincides with a reduction both in *svb* transcription and trichome number. Furthermore, they find that the addition of the full cis-regulatory region of *svb* on a different chromosome leads to preferential colocalization between the added region and the original mutated *svb* locus. The authors report that local concentration of Ubx in the vicinity of these colocalized enhancer regions recovers to wild-type levels under heat shock. They argue that, consistent with their initial hypothesis, the coincident partial recovery of *svb* transcription and trichome number indicates that this colocalization of multiple Ubx targets to a single microenvironment is indeed functioning to counteract the embryo's mutation-induced sensitivity to heat shock conditions.

Overall, the new experimental results presented in this work are deeply intriguing and hold the potential to offer significant new insights regarding the physical nature of transcription factor hubs, as well as their functional role in the spatiotemporal control of transcriptional activity. Nonetheless, several key questions remain regarding the interpretation of the experimental data and the proposed functional role of transcriptional microenvironments in facilitating phenotypic robustness in the face of genotypic and environmental perturbations.

Perhaps most fundamentally, the authors' conclusions hinge upon the assumption that the Ubx enrichment signal they observe at active loci play a causal role in modulating transcriptional activity, yet they present no data to directly support this assumption. Indeed, the box plots in Figure 3 C and D seem to show that, while both Ubx concentration and transcription decrease in response to heat, *DG3* deletion, or both, the relative impacts of these perturbations on transcription and Ubx concentration differ significantly in scale. This leaves open the possibility that the perturbations affect each feature separately, and that the correlation between transcription and local Ubx concentration are spurious, not causal. A simple but important check for this would be to show scatter plots of Ubx concentration vs. *svb* transcription within each condition. If the input concentration and output rate of transcription for a given locus are positively correlated when other conditions are held constant, this would constitute a solid basis for the authors' subsequent arguments. If, on the other hand, no correlation is evident, then this would fundamentally alter how the manuscripts' results are interpreted. This question can be resolved with data already in-hand and, in this reviewer's opinion, it will help clarify the significance of the reported experimental observations. Along similar lines, the strength of the work would benefit from the addition of plots that clarify not only how different variables (concentration, locus colocalization, transcription rate) change across conditions, but also with respect to one another.

On a more conceptual note, the authors' proposal that the colocalization of enhancers in transcription factor microenvironments could serve as an added layer of redundancy (akin to multiple TF binding sites and multiple enhancers) to increase the transcriptional robustness is thought-provoking. The concept of robustness is used throughout biology with different meanings, and further discussion of the implications and nuances of this proposal would also help establish what exactly the authors mean by it. For instance, what the authors cast as "robustness" could equally well be termed "desensitization". While robustness to environmental changes could be desirable, the fast pace of *Drosophila* development also necessitates rapid, precise transcriptional responses to changes in transcription factor concentrations. Thus, naively at least, if robustness leads to desensitization of the response of an enhancer to changes in the concentrations of inputs transcription factors, one might expect locus colocalization to actually be deleterious in the case of genes that must respond rapidly to time-varying transcription factor inputs. Augmenting the existing commentary to address this point would help clarify the implications of the present work. Does this trade-off make predictions regarding where and when this kind of clustering might manifest over the course of development (and for what genes)?

General Comments:

Can the authors comment on why the box plots in Figure 3 and Figure 4 contain so few data points? All contain 52 or fewer, yet one would expect there to be many more active *svb* loci per embryo at this point in development. If the full set of *svb* loci was not used for the analyses presented, can the authors comment on how the analysis subset was selected. Also, given that many of the effects presented are relatively subtle, the addition of data points (to the extent that it is feasible) would greatly enhance the robustness of the analysis and might lead to the discovery of additional trends within the data not evident in such small sample sizes. Given that the authors get multiple nuclei per embryos, is there a fundamental limitation to how much data they can present such as, for example, their data analysis pipeline?

As mentioned above, this work leans heavily upon the presumed causal relationship between local Ubx concentration and transcriptional output. The work would be greatly strengthened by the addition of bivariate analyses that rigorously test the relationship between Ubx concentration and transcription within each of the four conditions shown in Figure 3 C and D.

The boxplots in Figure 4C show that Ubx concentration at loci where the mutated *svb* site is colocalized with the *svbBAC* more or less recovers Ubx concentration at the endogenous (unperturbed) *svb* locus, yet the authors do not address whether the resulting signal is less than, equal to, or greater than the sum of the signal at each locus *(Df(X)svb* and *svbBAC*) when they are not colocalized. Showing how the enrichment signal at colocalized loci compares to the signal at each separately would indicate whether and to what degree some sort of synergy is at play, or whether the increased signal is merely a result of having more Ubx binding sites in region.

While the trends in Ubx intensity, radial separation, and transcriptional activity shown in Figure 4C, E, and F are interesting taken individually, much more could be learned from the data by making bivariate plots. How does transcriptional output at the mutated *svb* locus vary with the radial separation between loci? What about the Ubx intensity?

---

## [Author Response]

[Editors' note: the authors’ plan for revisions was approved and the authors made a formal revised submission.]

Essential points to address are the following:1) The connection between enhancers, Ubx protein concentration, transcriptional output, and phenotype. Although different concentrations of Ubx are measured at the deleted vs. wild-type locus, an important point that the three reviewers agree upon is that gene expression levels and Ubx concentrations are not well correlated. There appear to be other unexplained factors that are influencing the expression of svb that do not correlate with the TF concentration. Thus, it is unclear whether the experiments described here support or reject the hypothesis laid out in the "multi-enhancer "hubs" improve robustness by increasing transcription factor retention near transcription sites."

The reviewers are correct to point out that there are multiple inputs that control the response of the svb locus (Stern and Orgogozo, 2008). Therefore the response function of *svb* could have a positive but complicated relationship to Ubx concentrations and would depend on more than Ubx alone. We now state this in the second paragraph of the Discussion. However, Ubx is a crucial driver of *svb* and, specifically, also *DG3* expression in the ventral region of the A1 segment (Crocker et al., 2015) and Figure 1 of this manuscript. We specifically focused most of our analyses and quantifications in A1. We now state this in the Results section, “Transcription sites from the *DG3*‐ deletion allele have weaker Ubx microenvironment and lower transcriptional output”.

Within this specific body segment, Ubx intensity near *svb* transcription sites would be a reasonable metric of how local transcription factor concentrations changed when we perturbed the system through mutations and elevated temperature. We now state this in the Results section “Transcription sites from the *DG3*‐deletion allele have weaker Ubx microenvironment and lower transcriptional output”. With these caveats in mind, we did observe increased Ubx concentrations and around the wildtype *svb* transcription sites (Figure 3C, lower right panel) compared to the deletion allele, and increased phenotype resilience for the wildtype allele at 32 °C. In the *svbBAC* rescue, we also observed that colocalized *svbBAC* and *DG3*‐deleted *svb* allele had both higher Ubx concentrations and *svb* transcriptional outputs (Figure 4F). We, therefore, believe that our results support the idea that multi enhancer hubs help transcriptional factor retention and can ultimately lead to a more robust phenotype. We believe the analysis that the reviewers suggested below also supports our hypothesis of improved transcriptional factor retention with complexed enhancers.

The authors must consider further whether the global measurements shown in Figure 3 are the most effective measure of this proposed correlation, or whether a more fine-grained analysis might support the idea that boosted Ubx levels from complexed enhancers are driving transcription. As noted below, the authors might show scatter plots of Ubx concentration vs. svb transcription within each condition. A positive correlation would support their hypothesis.

We have modified the analysis in Figure 3C to show *svb* transcription vs. Ubx intensity. There is initially a positive correlation between svb transcriptional output and Ubx intensity, in line with the reviewers’ proposal. This trend dissipates at higher Ubx and *svb* intensities, indicating that the response of *svb* output to Ubx concentration is not a simple relationship. Capturing the exact dependence of transcriptional output on Ubx concentration is difficult for the reasons we stated in the second paragraph of the Discussion (e.g., multiple binding sites in enhancers and overlapping expression patterns). To fully address this would require future live imaging experiments that can track the gene locus regardless of its transcriptional state, in addition to reporting on its transcriptional activity and transcription factors around it. Numerous new reagents (fly lines, tagged proteins, etc.) are being developed in the lab to address this.

The trichome phenotype is also not directly correlated to the transcriptional outputs and Ubx concentrations, and as mentioned in the Discussion, further layers of regulation are likely in play. This idea needs to be more fully fleshed out.

We now have a new figure (Figure 6) summarizing our hypothesis and referred to this figure throughout the Discussion to clarify our proposed mechanism. In short, we propose in our revised Discussion that 1) the relationship between Ubx concentration and *svb* transcriptional output is positive but complex, likely involving additional factors (paragraph two) and 2) the relationship between *svb* transcriptional output and phenotype is sigmoidal (paragraph four) and the wild‐type system operates in a saturated regime under ideal conditions (paragraph five). The processes leading to the response functions in 1 and the phenotypical tolerance to a range of *svb* out in 2 remain to be investigated (paragraphs two and five), specifically with live imaging approaches. As mentioned in the previous paragraph, work is currently ongoing in the lab to address this.

A finding related to "enhancer hubs boosting local concentration" is the finding that an ectopic BAC appears not to influence Ubx concentration in a wild-type svb background, only in a deletion background. Does this finding indicate that such hub formation is saturable? And if so, does this indicate that the wild-type svb locus forms a local hub with enhancers in the immediate vicinity, and trans-complementation is not a normal feature of svb function?

We agree with the reviewers that hub formation is saturable, based on this observation. We also believe that local hub formation with the *cis*‐regulatory region of the wildtype svb is sufficiently in saturation to deal with environmental challenges as seen in trichome numbers. We now state this in the Discussion (paragraph five). Our data, however, does not provide a direct answer as to if trans‐ chromosomal interactions are a normal feature of *svb* or other genes. Ongoing efforts in the lab to characterize and map these long‐range interactions through imaging and genomics approaches should shed light on their functional impact during development in the future.

2) A second point raised concerned the exact cis regulatory regions in play. A minimal DG3 enhancer drives gene expression in ventral abdominal stripes, but does not rescue Ubx concentrations from a trans setting. A larger deletion that includes DG3 as well as additional Ubx binding regions (that are not sufficient for, but may be part of, ventral DG3-related activity) impacts transcription, trichome development, and robustness. A yet larger cis-regulatory domain on a BAC rescues some aspects of gene expression and Ubx concentration. The interpretation conflates DG3 with the function of the deleted region; reviewers noted that a more careful interpretation would differentiate results from each of these different cis elements. For instance, the lack of trans-rescue by the DG3 enhancer alone may be due to the inability of a short segment to transvect effectively. The interpretation should explicitly take into account known properties of transvecting regulatory loci in Drosophila.

We agree that we should take into account the efficiency of the rescue locus in finding the *svb* locus as an important factor in if phenotype rescue takes place. We now state that *DG3* alone as the rescue locus might have failed due to its inability to pair with the *svb* locus in the Discussion (paragraph four). We also now compare and contrast the *svbBAC*‐*svb* interactions that we observed with transvection and hypothesize that the addition of other topological elements such as insulator to *DG3* could overcome this problem (paragraph four). We further state that the *svbBAC* did not rescue trichomes in regions where *DG3* provided exclusive coverage, suggesting that the rescue BAC rescued the phenotype by overdriving the other ventral enhancers *E3* and *7* rather than directly restoring *DG3* function.

3) Several aspects can be addressed by better justification and explanation of methods and data presentation, including– Why sometimes either A1 or A2 trichomes are quantitatively assessed, depending on the figure;

This issue was an oversight on our part, and we have updated all the main figures where we counted trichomes to be from the A1 segment for consistency. We moved data from the A2 segment to figure supplements as they also show a similar but weaker trend as in A1. We speculate that this is due to additional factors at work as *DG3* in the A2 segment responds to additional inputs beyond Ubx, as explained in the Results section “The *DG3* enhancer responds specifically to Ubx in the A1 segment”.

– The use of one-tailed (vs. two-tailed) T-tests for statistical relevance;

We have changed our test to two‐tailed T‐tests throughout, as is the standard. This did not change of our findings.

– Recommendations for inclusion of both 25C and 32C phenotypes and data uniformly, and;

We added trichome images from 32 C to Figure 2 and for the data analysis involving Ubx intensity and svb transcriptional output in Figure 3. We did not conduct *svbBAC* rescue experiments at 25 C as neither the wildtype nor the deletion allele displayed reduced trichome numbers (phenotype output), and the difference between the Ubx intensities (molecular input) around transcription sites of both genotypes was small at this temperature.

– Clarification of exact role of Ubx in T1-T3 regulation, as the conclusions drawn about DG3 and Ubx roles are difficult to know based on the single images shown,

As the reviewers noticed, the role of Ubx in regulating *svb*, and *DG3* specifically, on the ventral surface in segments outside of A1 is more complicated (Figure 1B‐E). *DG3* expression in the thoracic segments T1‐T3 overlaps with other ventral *svb* enhancers and responds to changes in Ubx level less that in the A1 segment. T2 and T3 also show only low levels of expression from *DG3* on the ventral surface with wild‐type Ubx expression. This issue potentially introduces many confounding factors for quantitation. We have thus confined ourselves to qualitative descriptions of *DG3* properties outside of A1 and A2. We now state this concern at several places in the Results and Discussion.

– The number of data points in Figures 3 and 4 are limited; is there a technical limitation to more extensive sampling of the transcriptional readouts and Ubx intensities?

In the process of doing the additional analysis suggested by the review, we added more data points, and the number of transcription sites and embryos quantified is comparable to the original publication this manuscript is linked to (Tsai et al., 2017). As in the previous publication, colocalization is a relatively rare event, so we observed fewer transcription sites. The number observed for colocalized sites is also similar to our previous publication.

4) In some cases, the actual experimental approach was unclear to the reviewers:– There was some confusion about whether the DG3 deletion mutants were homozygous, so there would be no wild-type copy of the gene in these embryos – is that indeed the case?

Yes, we selected larvae or embryos homozygous for the *Df(X)svb^108^*allele based on the following phenotype in the T1 segment: the lack of trichomes or the lack of *svb* mRNA expression, respectively. This is a homozygous marker for the deletion allele as the wild‐type *svb* allele expresses in this segment. This was mentioned in “Deletion of a region including *DG3* enhancer causes defects in ventral trichome formation specifically at elevated temperatures” in Results, “Preparing *Drosophila* embryos for staining and cuticle preps” in Materials and methods and the legends for Figure 2C in our original manuscript (2E in the revised version,). We have now made it explicit in the Results that we use this to select for animals homozygous for the deletion for both counting trichome number and for confocal imaging. It is now also described in both “Cuticle preparations and trichome counting” and “Imaging fixed embryos” in the Materials and methods section.

– Figure 4 shows svb, but not dsRed mRNA expression; is that correct? Imaging of dsRed only used to score a svb locus as "overlapped" vs. "not overlapped"?

Yes, the data analysis in Figure 4 only used only *svb* mRNA output. Imaging of *dsRed* transcription site is only used to score colocalization with *svb*. We displayed image panels for individual nuclei that show a *dsRed* signal for illustrative purposes. We explicitly stated that we are displaying *svb* transcription in both the Figure 4E and the legends.

– Choice of pixel size for ROI.

The 40‐pixel ROI is too large and a mistake in our part. The actual ROI size in the previous manuscript was a 4‐pixel square ROI; this has been changed to be a circle with a diameter of 4 pixels (170 nm). We chose this size because it is the resolution limit of the AiryScan images we acquired. We have corrected this and stated our rationale in the revision when we describe the image analysis in “Transcription sites from the *DG3*‐deletion allele have weaker Ubx microenvironment and lower transcriptional output” in the Results and “Analysis of microenvironment and *svb* transcription intensity” in the Materials and methods.

– Signal to noise for Ubx across the nucleus, as well as variation in average Ubx from sample to sample.

We used exactly the same antibodies and methodology as reported in (Tsai et al., 2017) to stain for Ubx and the quality of the images acquired were the same. In the previous publication we explored the Ubx cluster characteristics. We now include a new supplemental figure (Figure 3—figure supplement 1) with Ubx variation across the nucleus within embryos, and from sample to sample (there is no significant difference). In contrast, at the sites of active *svb* transcription there is a significant difference (p < 0.001, two‐tailed t‐test) from randomly sampled locations in the nucleus. This has also been added to the Results “Transcription sites from the *DG3*‐deletion allele have weaker Ubx microenvironment and lower transcriptional output”